# New endophytic strains of *Trichoderma* promote growth and reduce clubroot severity of rapeseed (*Brassica napus*)

**Mahmodol Hasan[1]\*, Motaher Hossain📷[2]\*, Daohong Jiang[3]**

**1** Plant Pathology Laboratory, Department of Agronomy and Agricultural Extension, University of Rajshahi, Rajshahi, Bangladesh, **2** Department of Plant Pathology, Bangabandhu Sheikh Mujibur Rahman Agricultural University, Gazipur, Bangladesh, **3** College of Plant Science and Technology, Huazhong Agricultural University, Wuhan, PR China

\* mahmudul742001@yahoo.com (MH); hossainmm@bsmrau.edu.bd (MH)

**Data Availability Statement:** All relevant data are within the paper and its Supporting Information files.

**Funding:** The author(s) received no specific funding for this work.

## Abstract

Rapeseed (*Brassica napus* L.) is the world's third most important edible oilseed crop after soybean and palm. The clubroot disease caused by *Plasmodiophora brassicae* poses a significant risk and causes substantial yield losses in rapeseed. In this study, 13 endophytic fungal strains were isolated from the healthy roots of rapeseed (*B. napus*) grown in a clubroot-infested field and molecularly identified. Based on germination inhibition of resting spores of *P. brassicae*, two endophytic fungal antagonists, *Trichoderma* spp. ReTk1 and ReTv2 were selected to evaluate their potential for plant growth promotion and biocontrol of *P. brassicae*. The *Trichoderma* isolates were applied as a soil drench ($1 \times 10^7$ spore/g soil) to a planting mix and field soil, in which plants were grown under non-infested and *P. brassicae*-infested ($2 \times 10^6$ spore/g soil) conditions. The endophytic fungi were able to promote plant growth, significantly increasing shoot and root length, leaf diameter, and biomass production (shoots and root weight) both in the absence or presence of *P. brassicae*. The single and dual treatments with the endophytes were equally effective in significantly decreasing the root-hair infection, root index, and clubroot severity index. Both ReTk1 and ReTv2 inhibited the germination of resting spores of *P. brassicae* in root exudates. Moreover, the endophytic fungi colonized the roots of rapeseed extensively and possibly induced host resistance by up-regulated expression of defense-related genes involved in jasmonate (*BnOPR2*), ethylene (*BnACO* and *BnSAM3*), phenylpropanoid (*BnOPCL* and *BnCCR*), auxin (*BnAAO1*) and salicylic acid (*BnPR2*) pathways. Based on these findings, it is evident that the rapeseed root endophytes *Trichoderma* spp. ReTk1 and ReTv2 could suppress the gall formation on rapeseed roots via antibiosis, induced systemic resistance (ISR), and/or systemic acquired resistance (SAR). According to our knowledge, this is the first report of the endophytic *Trichoderma* spp. isolated from root tissues of healthy rapeseed plants (*B. napus*.), promoting plant growth and reducing clubroot severity.

**Competing interests:** The authors have declared that no competing interests exist.

## Introduction

Rapeseed (*Brassica napus* L.) is one of the most important culinary oilseed crops predominantly grown in most of Asia and Europe, New Zealand and Canada. It is the third major source of vegetable oil in the world after soybeans and palm and is also becoming an increasingly popular ingredient in bio-diesel production [1]. Clubroot, caused by the obligate parasite *Plasmodiophora brassicae* Woronin, is one of the most severe soilborne diseases of rapeseed (*B. napus*) and other cruciferous plants worldwide [2]. It is gradually becoming a threat to rapeseed production in many countries like Canada, New Zealand, China, and the Philippines [3–5]. The increase in farming areas and proximal crop rotation are factors in the rise in clubroot [2]. *P. brassicae* is an obligate biotrophic pathogen that can cause intracellular infection in the roots. When rapeseed roots become infected, the root cells begin to grow abnormally, forming lumps that resemble tumours [6]. Clubroot infection prevents the normal absorption of water and nutrients, resulting in loss of vitality and stunting of the plant. The damage in oilseed rape ranges from 10% to a complete yield loss, including plant and seed losses [7]. Recent works have extensively described the disease development and cellular alterations in the host plants after infection [8, 9]. The life cycle of the pathogen is complex and comprises two significant phases. In the first phase, dormant spores of *P. brassicae* penetrate root hairs and epidermal cells, forming primary plasmodia. In the second phase, primary plasmodia of *P. brassicae* release secondary zoospores, which penetrate the root cortex and form galls. The fully developed secondary plasmodia in the root cortex can produce thick-walled dormant spores, which survive for many years without a host in soil [10], making the disease difficult to control. Chemical control is not necessarily practical or economically feasible to manage the disease. Therefore, exploring other sustainable ways to tackle this disease efficiently is necessary.

Applying biological control strategies could particularly aid in controlling club root pathogen [11]. Biocontrol agents that have been explored against clubroot pathogen include endophytic bacteria or fungi [11, 12]. Endophytic microorganisms grow and develop entirely within their host plant tissues [13]. Possible plant benefits of endophytic colonization include increased plant growth, improved tolerance to abiotic stresses and reduced pest and diseases [13–16]. Among these beneficial microorganisms, the fungal genus *Trichoderma* has been extensively studied as a plant growth promoter, inducer of abiotic stresses and biological control agent of plant pathogens in many plant species, including rapeseed [17–23]. For instance, inoculating rapeseed plants with *Trichoderma atroviride* enhanced the growth characteristics and yield of two rapeseed cultivars, particularly under high levels of copper in the soil [24]. The application of *Trichoderma harzianum* increased seed weight and played a key role in the induction of systemic defense against the fungal pathogen in rapeseed plants [25]. The concurrent application of *Trichoderma harzianum* and an arbuscular mycorrhizal fungus to Arabidopsis and rapeseed roots resulted in a significant increase in colonization by both fungi and enhanced productivity in both Brassicaceae species [26]. *Trichoderma parareesei* favours the tolerance of rapeseed to salinity and drought [27]. In several studies, *Trichoderma* strains have been proven effective as biological control agents for *P. brassicae* in cruciferous and Brassicaceae plant species. Experiments conducted in greenhouse pots revealed that *T. harzianum* strain T4 had a control efficiency of around 79% against *P. brassicae* in Chinese cabbage [28]. Likewise, the antifungal properties of the *T. harzianum* strain LTR-2 were demonstrated in Chinese cabbage grown in the field. The incidence of the disease was reduced from 96.7% (untreated control) to 51.3% (seeds treated with spores of *T. harzianum* LTR-2) [29]. Another study has demonstrated the biocontrol effect of two strains of *Trichoderma* against *P. brassicae* in Arabidopsis and rapeseed, with the Hz36 strain having the maximum biocontrol efficiency on clubroot for rapeseed (44.29%) and Arabidopsis (52.18%), respectively [30]. The disease

reduction occurs without risking the environment, demonstrating that utilizing *Trichoderma* is beneficial in managing clubroot.

Biocontrol by *Trichoderma* against fungal phytopathogens is achieved through a number of direct and indirect mechanisms such as mycoparasitism, competition for resources and space, modification of microbial community, promotion of plant growth, induction of plant defensive mechanisms and antibiosis [31–35]. *Trichoderma* can produce secondary metabolites that could significantly inhibit the germination of resting spores of *P. brassicae* [30]. Release of elicitors by *Trichoderma* may transduce various signals within the plant, such as salicylic acid (SA), jasmonic acid (JA), or ethylene (ET), leading to the elicitation of the plant defense responses against the pathogen [32–34]. These mechanisms are part of a multiple-component action exerted by *Trichoderma* to achieve effective biocontrol under various environmental conditions. Although *Trichoderma* encompasses several endophytic species, a few have been investigated to improve clubroot management. Therefore, the present work was undertaken to identify endophytic *Trichoderma* antagonists to control *P. brassicae*, the causal agent of clubroot of rapeseed. Moreover, their ability to colonize roots and induce defense responses in rapeseed was also explored. This study aimed to locate new prospective candidates with a deeper understanding of their mode of action for use as clubroot biocontrol in rapeseed.

## Materials and methods

### Isolation and identification of endophytic fungi from roots of rapeseed

To isolate the endophytic fungi, healthy rapeseed root samples were collected from a clubroot infested field in Zhejiang, Hubei province, PR China. Additional clubbed root samples were also collected from the same field to extract resting spores of *P. brassicae* for subsequent use in this experiment. The collected healthy roots were washed in tap water to remove any soil particles, then immersed in 75% ethanol (vol/vol) for 60 sec, followed by 5 min dipping in 2.5% NaOCl (vol/vol), and finally rinsed three times in sterile distilled water. Twenty surface-sterilized roots (young and old) were aseptically cut into 0.5 cm lengths (avoiding root tips) for a total of 285 segments and transferred to potato dextrose agar (PDA) plates (three pieces in each plate) supplemented with Cef (50 μg/ml) to prevent bacterial growth. A total of 95 plates were sealed with parafilm to avoid desiccation. Plates were incubated at 25˚C for 14 days, and hyphae developing from the segments were observed regularly. The fungi growing on root segments were then transferred to fresh PDA, and finally, the isolates were purified by single spore culture. To purify non-sporulating fungal isolates, hyphal tip culture was performed. The fungal isolates were initially identified based on cultural and morphological characteristics, *viz*. colony pigments, conidial morphology, and conidiophores structures.

### Identification with fungal ITS amplification and sequencing

Twenty-five fungal isolates were grown on cellophane membranes placed on the PDA plates, and DNA was extracted according to the standard procedure [36]. The fungal ribosomal DNA (rDNA) internal transcribed spacer (ITS) regions 1 and 2 were amplified by PCR using primers ITS1-F (5′–TCCGTAGGTGAACCTGCGG–3′) and ITS4-R (5′–TCCTCCGCTTATTGAT ATGC–3′) [37]. The reaction products were separated in 1.0% (wt/vol) agarose gel, and the amplicons were purified using a gel band purification kit (Axygen Incorporation, China). The PCR product was then ligated into a pMD18-T vector and transformed into *Escherichia coli* JM109 (Promega) according to the manufacturer's instructions, resulting in clone libraries. The transformants were plated on LB agar plates containing 50 μg/ml kanamycin, streptomycin, and rifampicin. Finally, the randomly selected positive clones were sequenced in an ABI 3730 sequencer (Applied Biosystems, United States) using the Primer M13F. The final

sequence sets were submitted to BLAST analysis and compared to the most closely related strains. The endophytes were considered conspecific at a threshold identity of ≥99%. Therefore, 13 isolates were identified up to genus based on their sequence identity. To verify the phylogenetic position of *Trichoderma* genotypes, sequences of candidates isolates and the corresponding best BLAST hits were aligned by Clustal X and manually corrected in GENE-DOC. Neighbor-joining (NJ) tree was built with the MEGA 11 software package using a Kimura two-parameter (K2P) model [38], and the stability of clades was tested using 1,000 bootstrap replications.

## Selection of endophytes for biocontrol activity

To identify fungal endophytes with biocontrol activity against the pathogen, isolated endophytes were tested for their ability to inhibit the germination of *P. brassicae* resting spores, as described in the later section "Effect of *Trichoderma* on the resting spore germination of *P. brassicae*" of this manuscript. Moreover, the temperature requirements of the fungi were determined by monitoring their growth at 15, 20, 25, 30, and 35˚C.

## Effect of *Trichoderma* on growth promotion of rapeseed

Two sets of experiments were performed, one with field soil and another with soilless planting mix, to evaluate the growth promotion potentials of each *Trichoderma* isolate. Plastic pots (10.5 cm × 9.0 cm) were filled separately with soilless planting mix and field soil. Before seeding, each pot was inoculated with either ReTk1 or ReTv2 by applying 25 mL ($1 \times 10^7$ conidia/mL) of their spore suspension to the potting media. Followingly, *P. brassicae* spores were added to the potting media at a rate of $1 \times 10^7$ spores per gram of medium. Immediately after inoculation, 4 seeds of rapeseed were sown in each pot at equidistance from each other. Plants grown without inoculation with *Trichoderma* were regarded as controls. The experiment used a randomized complete block design (RCBD) with individual trials as the blocking factor. Each treatment consisted of 60 plants, and each trial was replicated thrice. After seeding, the pots were arranged randomly on a holding rack in a greenhouse at 20–23˚C with a 14-h photoperiod (512 µmol m$^{-2}$ s$^{-1}$). At the end of the experiment (42 days after seeding), plant growth parameters such as first true leaf diameter (25 days after seeding), shoot length, root length, shoot weight, and root weight were recorded. Similar experiments were conducted with the model plant *Arabidopsis thaliana* using a soilless planting mix as a growth medium.

## Effect of *Trichoderma* on the resting spore germination of *P. brassicae*

Clubroot galls were collected from infested rapeseed fields of Hubei province to prepare resting spores as described above. After removing soil particles by washing, the galls were air-dried and stored at -80˚C until required. To extract resting spores, about 5 g of dried galls were soaked in 150 mL distilled water for 2 h to soften the tissue and then macerated in a blender machine at high speed for 2 min. The slurry was filtered through eight layers of cheesecloth, and the spore concentration was approximated using a hemocytometer. The crude resting spore suspension was used for the inoculation of soil. On the other hand, a part of the resting spores of *P. brassicae* was purified by gradient centrifugation as described by Suzuki *et al.* [39] with minor modification and adjusted to $10^7$ spores/ ml with sterile distilled water.

   To collect root exudates, seeds of rapeseed were germinated by incubation on moistened filter paper for three days. Three-day-old seedlings were transferred to Hoagland solution in plastic cups (150 ml solution). The seedlings were grown in a growth chamber at 22˚C for 14 days. After the removal of seedlings from the cups, root exudate solution (RES) was collected, filter-sterilized using a biofilter (pore size 0.22 µm), and kept at 4˚C until use.

To prepare culture filtrates of *Trichoderma* spp., flasks containing PDB (250 mL) were inoculated with agar discs of ReTk1 and ReTv2 and incubated on rotary shakers at 180 rpm and 25°C for 6 days. After centrifuging at 4000 rpm for 15 min, the culture filtrates were collected.

To assay the effect of *Trichoderma* ReTk1 and ReTv2 on the inhibition of germination of the resting spores of *P. brassicae*, 5 ml of root exudates, 0.5 ml purified resting spore suspension and 1.0 ml culture filtrate from ReTk1 and ReTv2 were added to a sterile tube and the pH was adjusted to 6.3, with three replications. Culture filtrate replaced by an equal amount of PDB was treated as a control. The tubes were incubated at 24°C for up to 6 days in darkness. The microscopic examination of resting spore germination was done at 2, 4, and 6 days by staining with 1% orcein-acetic acid, according to Naiki et al. [40]. The germinated spores were considered empty under a light microscope, in contrast to red non-germinated spores.

## Effect of *Trichoderma* spp. on root hair infection and clubroot severity caused by *P. brassicae*

The experiment was conducted using field soil and soilless planting mix in the greenhouse. Plastic pots (10.5 cm /9.0 cm) were filled with field soil and soilless planting mix (pH 5.5–7.5; Peilei Technology Development Company, Zhen Jiang, China), which were later saturated with acidified water (2 M HCl) to adjust the pH at 6.3. Two days before seeding, the potting medium in each pot was inoculated with 5 mL of a *P. brassicae* suspension ($1 \times 10^7$ resting spores /mL) to produce an infestation level of about $2 \times 10^6$ spores/g of growth medium. The conidial suspensions (25 mL, $1 \times 10^7$ conidia mL$^{-1}$) of ReTk1 and ReTv2 were applied as a soil drench immediately after seeding of rapeseed (Cv. Hua you za 62) seeds (4 seeds/pot). Control treatment received 25 mL water only. The experiment was conducted following randomised complete block design (RCBD) with individual trials as the blocking factor. Seeded pots were arranged randomly on a holding rack and placed in a greenhouse at 20–23°C with a 14-h photoperiod (512 μmol m$^{-2}$ s$^{-1}$). Each treatment consisted of 60 plants in a trial, and each trial was replicated at three different times, with all materials prepared each time independently.

Roots of randomly sampled three treated plants were uprooted 7 and 14 days after seeding (DAS), washed thoroughly to remove all soil particles, rinsed three times in sterile distilled water, cut into 1 cm pieces, and fixed in 70% ethanol until use. Roots of untreated control plants (pathogen-inoculated only) were processed similarly. Treatment-wise, the root pieces were stained overnight with aceto-carmine (1%) and observation was done using a light microscope for the presence of primary plasmodia or zoosporangia in root hairs. Up to 40 root hairs were checked for each plant, and the incidence of infection was calculated.

The rest of the plants were maintained in the greenhouse for 6 weeks and watered as necessary. At the end of the experiment (42 DAS), disease severity was assessed according to Klewer et al. [41] using a 0–4 scale, where 0 = no symptoms, 1 = very small clubs, mainly on lateral roots that do not impair the main root, 2 = small clubs covering the main root and few lateral roots, 3 = medium-sized to bigger clubs, also including the main root, plant growth might be impaired, 4 = severe clubs in lateral, main root or rosette, fine roots were completely destroyed, plant growth was affected. The disease severity index (DSI) was calculated using the formula: DSI = $(1n_1 + 2n_2 + 3n_3 + 4n_4)$ x $100/4N_t$, where $n_1$–$n_4$ is the number of plants in the indicated class and $N_t$ is the total number of plants tested.

## Application timing of *Trichoderma* spp. targeting primary and secondary zoospores of *P. brassicae*

An experiment was conducted to determine the optimal timing for applying ReTk1 and ReTv2 to target primary and secondary zoospores of *P. brassicae*. The detailed treatment layout is

shown in Table 3. ReTk1 and ReTv2 or water (control) were applied as a soil drench to pathogen-infested field soil and planting mix at seeding time. Seedlings were uprooted 7 and 14 DAS, and, after rigorous washing, transplanted into a non-infested medium treated with ReTk1 and ReTv2 or water only (control). The effect of *Trichoderma* spp. on infection caused by primary and secondary zoospores was determined. This experimental plan was based on the peak of root-hair infection (RHI) and release of secondary zoospores under optimal temperature conditions, as reported by Sharma et al. [42].

## Examination of rapeseed-root colonization by *Trichoderma* spp.

**Transformation of *rfp* gene into *Trichoderma* spp..** Strain EHA105 of *Agrobacterium tumefaciens* and binary vector pCHMC (promoter EF-1α) with a backbone of pCAM-BIA1301 and PtrpC::hph from vector pSKH were used for ReTk1 & ReTv2 transformation. The method used for transformation was followed by Qin et al. [43] with some modifications. ReTk1 and ReTv2 were cultured in PDA slants for 5–7 days, and then the colony surface was washed with sterile double-distilled $H_2O$ (dd$H_2O$). The spore suspension was passed through two layers of cheesecloth to remove mycelial debris and then centrifuged at 4400 rpm for 10 min to pellet the spores. The pellets were stored at 4°C for less than 1 h. To co-cultivate *Trichoderma* spp. (ReTk1 and ReTv2) and *A. tumefaciens*, spore pellets of ReTk1 and ReTv2 were re-suspended with prepared bacterial cell suspension. The spore concentration was adjusted to $1 \times 10^7$ spores/ml. An aliquot of 200 μl of spore-bacterial cell mixture was transferred and spread onto a cellophane membrane covering the comedium (Co-IM) containing 400 μM of AS. After incubation at room temperature for 2 days in dark conditions, the membrane was transferred to a sterile empty plate. Then it was covered with 20 ml of cooled PDA containing 500 ug/ml of cephalosporin to counter-select bacteria and 50 ug/ml of hygromycin to select *Trichoderma* transformants primarily. After incubation at 20–22°C for 3–4 days, the candidates were transferred to fresh PDA plates containing 100 ug/ml of hygromycin for a second round of selection. Transformant strains that expressed *rfp* and showed the same physiological and morphological characteristics as the wild-type in culture media were isolated, and two of the transformants, ReTktr-1(5) and ReTvtr-2(23), were chosen for CLSM analysis.

## Inoculation of transformants into rapeseed root and microscopy

The transformants ReTktr-1(5) and ReTvtr-2 (23) germlings were obtained by inoculating $10^6$ conidia/ml in 200 ml of potato dextrose broth (PDB) (Sigma, Madrid, Spain) and then further incubated for 15 h at 28°C at 200 rpm. The germinated conidia were centrifuged at 4,400 rpm for 10 min, washed three times with sterile distilled water, and diluted to $10^5$ germlings/ml. Bunches of 10 rapeseed seedlings grown on MS media for 7 days were tied together and placed inside 250-ml flasks containing 150 ml of salt minimal medium MM [44] supplemented with 0.2% (w/v) glycerol as the sole carbon source and 20 mg ammonium sulfate/l. The roots of the rapeseed seedlings were submerged in the medium up to the hypocotyls. The flasks were then inoculated to a final concentration of $10^5$ germlings/ml and incubated for 10–24 h at 22°C with shaking at 115 rpm for colonization studies. The same experiment but without inoculation of the fungus was treated as a control. Rapeseed roots colonized by transformants (ReTk1 & ReTv2) were taken from the hydroponic cultures after 10 and 24 h incubation, thoroughly washed with distilled water and microscopy was carried out in three independent experiments. A confocal lesser scanning microscope (CLSM) with a Leica TCS SP2 was used to study the plant-fungus interaction *in vivo*. The excitation wavelength was 488 nm (argon/krypton laser) and the emission wavelengths

were 500–550 nm for RFP. A dichroic filter RSP 500 was also used. A panoramic view was obtained with a ×20 objective and detailed views with ×40 and ×63 objectives and a zoom factor. Images were acquired by Leica Confocal Software 2.5.

### Relative expression of defence-related genes in rapeseed plants

At 14 DAS, roots from *Trichoderma*-treated rapeseed plants were removed and washed with running tap water and then rinsed in sterile distilled several times. RNA was extracted from the collected roots using the TRIzol reagent (Invitrogen, USA). RNA extracted from root samples of non-treated plants was used as a control. Contamination of DNA was removed by DNaseI treatment (RNase free) (TaKaRa, Dalian, China) according to the manufacturer's instructions. Five micrograms of total RNA (free of DNA) were converted to cDNA using the RevertAidTM First-strand cDNA Synthesis kit (MBI, Fermentas, USA) following the manufacturer's instructions. To examine the relative expression of plant defence-related genes involved in JA (*BnOPR2*), ET (*BnACO* and *BnSAM3*), phenylpropanoid (*BnOPCL* and *BnCCR*), auxin (*BnAAO1*) and SA (*BnPR1*, *BnPR2* and *BnPR5*) pathways, qPCR was done using gene-specific primers (S1 Table in S1 File). An aliquot of a 1 μL cDNA sample, 10 μL SYBR Green I, 0.5 μL (50 nM) of each primer and 8 μL DEPC-treated water were brought to a total reaction volume of 20 μL. PCR amplification was carried out for each target gene on CFX96[TM] Real-Time System (Bio-Rad, USA) using previously reported PCR conditions [45, 46]. The relative expression ratio of a target gene was measured using STEPONE v. 2.1 software (Applied Biosystems) based on the differences in Ct of a sample versus the control. Differences between a treatment and the control in the relative expression of each gene were assessed for significance at $P < 0.05$.

### Data analysis

All analyses were conducted in duplicate, with three replicates for each experiment. Data are expressed as the mean ± standard error. The results were analyzed using the SPSS statistical package (SPSS, Chicago, IL, USA). An analysis of variance (ANOVA) was used to evaluate the treatments. When a significant treatment effect was found ($P = 0.05$), Duncan's Multiple Range Test was used to determine the significant difference at $P \leq 0.05$.

## Results

### Isolation and identification of endophytic fungi

A total of 25 isolates of endophytic fungi were recovered and purified from healthy and surface-sterilised root tissues of rapeseed. Some fungi readily sporulated on the PDA medium after 5–6 days of incubation in darkness at 25˚C, while others did not produce any spores or conidia. Thirteen sporulating isolates were identified up to genus and/or species level based on their conidial morphology, conidiophore structure and other unique phenotypic characteristics. Finally, molecular identification was done based on rDNA ITS sequence analysis to confirm the reliability of morphological identification. Using NCBI BLAST, 13 different fungal genotypes with > 90% sequence similarity with the putative taxonomic affinity were identified (Table 1). One endophytic fungal isolate shared a 100% identical rDNA sequence with a previously uncultured organism.

Among the 13 fungal endophytes, isolates ReTk1 and ReTv2 were selected for biocontrol activity against *P. brassicae* as these two strains showed the highest inhibition (over 50% efficacy) of germination of the resting spores of *P. brassicae*. Moreover, the selected two isolates exhibited broad temperature requirements and abundant conidia production ability. The

**Table 1. Closest rDNA sequence matches (BLASTN) of the endophytic fungal isolates.**

| Representative isolate code | Putative taxonomic affinity | Similarity (%) | Score (Expected value) | Gene bank accession number |
|---|---|---|---|---|
| ReR1 (Red) | *Talaromyces amestolkiae* | 100 | 1075 (0.0) | OQ975662 |
| ReY1 | *Mucor hiemalis* | 99 | 1173 (0.0) | OQ975663 |
| ReBWF | *Pythium* sp. | 99 | 1080 (0.0) | OQ975664 |
| ReWCF | *Pythium spinosum* | 100 | 617 (Ie-173) | OQ975665 |
| ReBF1 | *Alternaria altenata* | 99 | 804 (0.0) | OQ975666 |
| ReWFL | *Fusarium oxysporum* | 99 | 1026 (0.0) | OQ975667 |
| ReP1 | *Penicillium* sp. | 100 | 1096 (0.0) | OQ975668 |
| ReFU | *Fusarium tricinctum* | 100 | 547 (Ie-152) | OQ975669 |
| ReScl | *Botryotinia fuckeliana* | 100 | 547 (Ie-152) | OQ975670 |
| ReP3 | *Sclerotinia sclerotiorum* | 91 | 721 (0.0) | OQ975671 |
| ReF4 | *Alternaria* sp. | 100 | 1064 | OQ975672 |
| ReTv2 | *Trichoderma* sp. *atroviride* | 99 | 1098 (0.0) | OQ975673 |
| ReTk1 | *Trichoderma* sp. *koningiopsis* | 99 | 1110 (0.0) | OQ975674 |

optimal temperature range for endophytes was between 20 and 30˚C, while the fungi grew well over the temperature range of 15 to 35˚C, indicating that both fungi are easily adaptable as biocontrol agents over a wide temperature range (S1A, S1B Fig in S1 File). Both strains produced septate, hyaline, and smooth-walled vegetative hyphae. The culture on PDA of ReTk1 strain was thick, dense, green without forming a concentric ring. On the other hand, the strain ReTv26 had a sparse and light culture. Conidia of both strains were one-celled, globose, smooth-walled, pale green forming on hyaline, smooth-walled branched and verticillate conidiophores morphological traits, the two selected strains were classified as *Trichoderma*. Sequence results obtained revealed that isolate ReTk1 and ReTv2 showed 99% sequence similarity with *T. koningiopsis* and *T. atroviride*, respectively. However, ITS sequences alone are insufficient for species-level identification of fungi, particularly *Trichoderma* strains, because multiple species share the same ITS sequence. Identification of *Trichoderma* strains at the species level should be aided by using additional loci such as *rpb2* and *tef1* [47]. Thus, ReTk1 and ReTv2 have been recognized as unconfirmed species of *Trichoderma* and described as *Trichoderma* spp. ReTk1 and ReTv2 in the subsequent sections of the manuscript.

## The root endophytic *Trichoderma* spp. promoted the growth of rapeseed

To assess the effect of TReTk1 and ReTv2 on plant growth, true leaf diameter was measured at 25 DAS, while the shoot length, root length, shoot weight, and root weight was evaluated at 42 DAS in comparison to the control. Both isolates significantly enhanced the leaf diameter of plants grown in field soil and planting mix compared to the untreated control. Similarly, TReTk1 and ReTv2-treated plants had significantly much longer root and shoot lengths than non-treated plants. In both field soil and planting mix trials, TReTk1 and ReTv2 inoculation significantly enhanced shoot and root biomass, resulting in a significantly higher fresh shoot and root weight than untreated controls (Fig 1).

## Inhibition of *Trichoderma* spp. on the resting spore germination of *P. brassicae*

At the beginning of the spore reaction (0 days), the germination percentage of resting spores of *P. brassicae* was zero in all treatments. Two days after incubation, the proportion of germinated spores was higher in the controls than in the treatments with culture filtrates of ReTk1

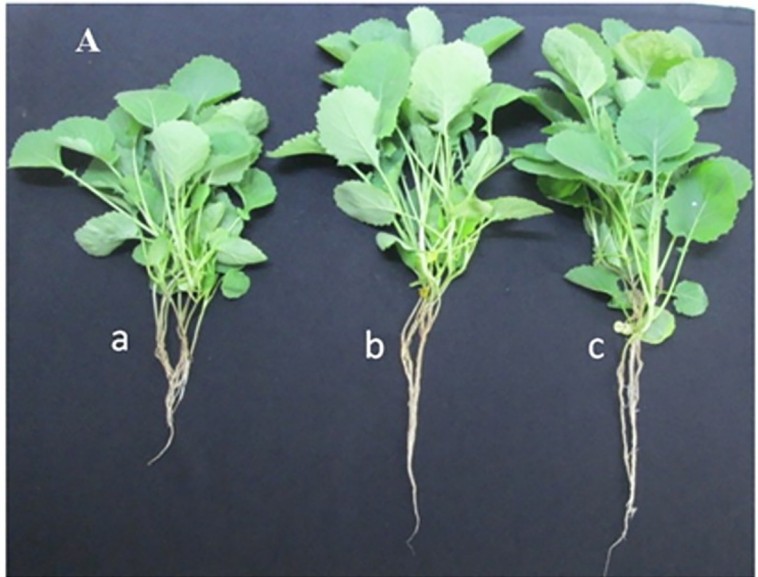

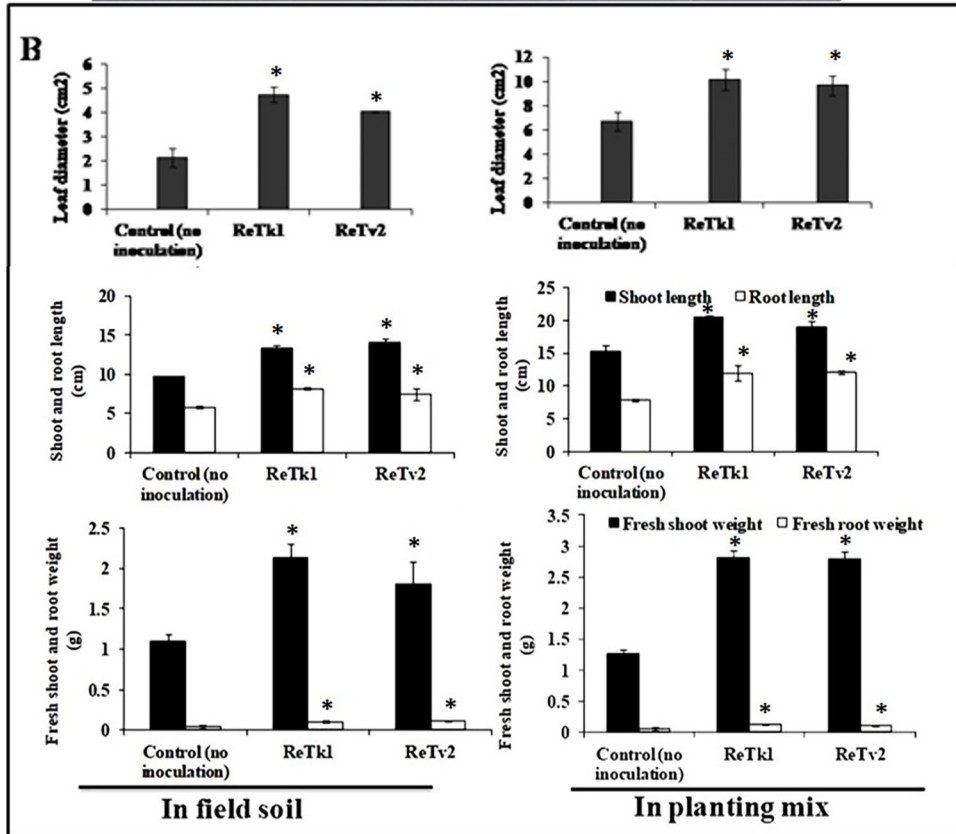

**Fig 1. Effect of endophytic fungi *Trichoderma* spp. ReTk1 and ReTv2 inoculation on growth promotion of rapeseed.** The experiment was carried out under greenhouse conditions. Field soil and planting mix were inoculated with ReTk1 and ReTv2 ($1 \times 10^7$ conidia/gm soil) by soil drench method. (A) The phenotypes of rapeseed plants without treatment (a) or treatment with ReTk1 (b) and ReTv2 (c). (B) The growth promotion of rapeseed plants. The proliferation of the first true leaf diameter was measured 25 days after seeding and other growth parameters were quantified 42 days after seeding. The mean average of six replications and capped lines represent standard error. The symbol "*" on top of the bar indicates a significant difference compared to the control at $P<0.05$.

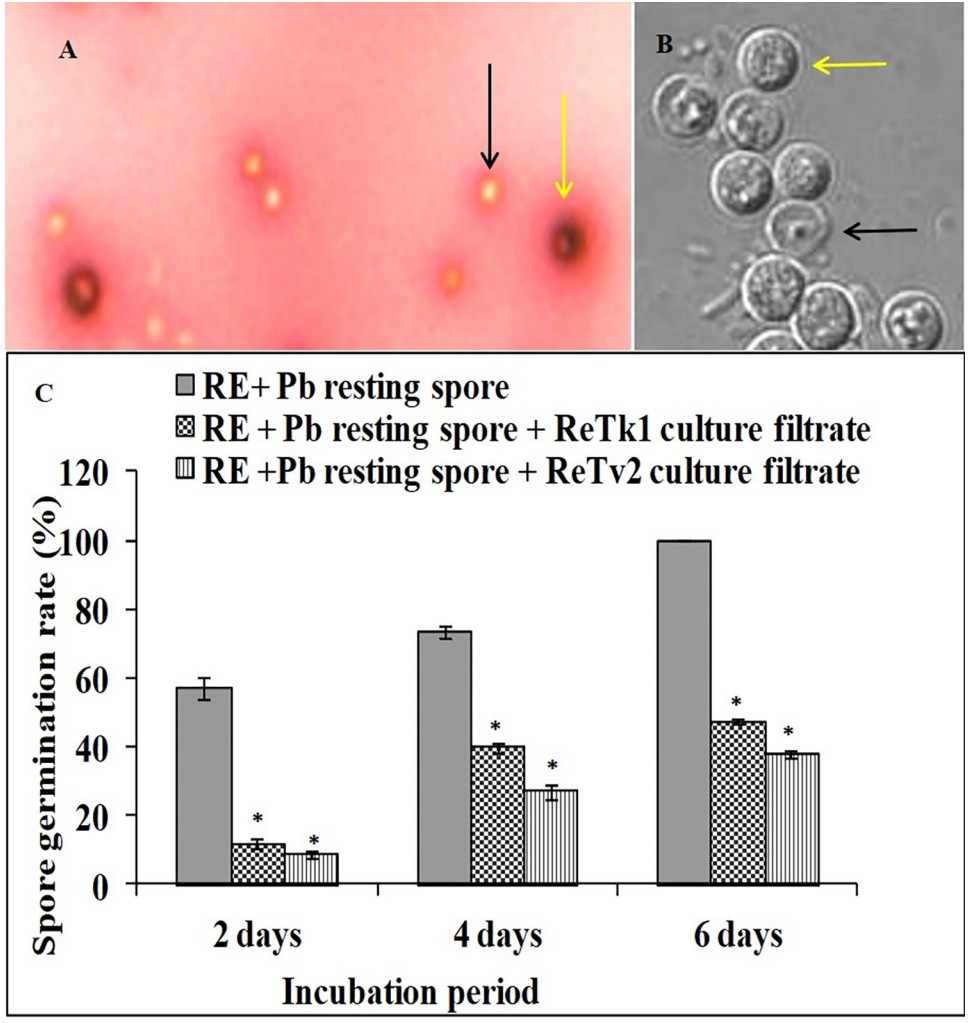

**Fig 2. Inhibition of endophytic fungi *Trichoderma* spp. ReTk1 and ReTv2 on resting spore germination of *P. brassicae* (Pb) in root exudate solution (RES).** RES amended with culture filtrates of ReTk1 & ReTv2 over six days incubation period. The germination of resting spores was counted by staining with orcein-acetic acid. Germinated spores were empty under light microscope, in contrast to red non-germinated spores (A). The germination also checked under CLSM (B). The black and yellow arrowheads in the figure indicate germinated and non-germinated resting spores, respectively. (C) The mean average of six replications and capped lines represent standard error. The symbol "*" on top of the bar indicates a significant difference compared to the control at *P<0.05*.

and ReTv2. After four days, 73% spore germination was observed in the control treatments, whereas, ReTk1 and ReTv2 treatments displayed only 39.74 and 26.87% germination of the resting spores. Following six days of incubation, 100% of the resting spores germinated in the control treatments, compared to only 47.31 and 37.57% in the ReTk1 and ReTv2 treatments, respectively (Fig 2).

### *Trichoderma* spp. suppressed clubroot severity of rapeseed

The biocontrol effect of endophytic fungi *Trichoderma* spp. ReTk1 and ReTv2 on clubroot severity of rapeseed was evaluated in field soil and planting mix experiments. The percentage of root-hair infection (RHI) by *P. brassicae* was significantly reduced in plants grown in field soil and planting mix treated with ReTk1 and ReTv2 at 7 and 14 DAS, respectively, compared

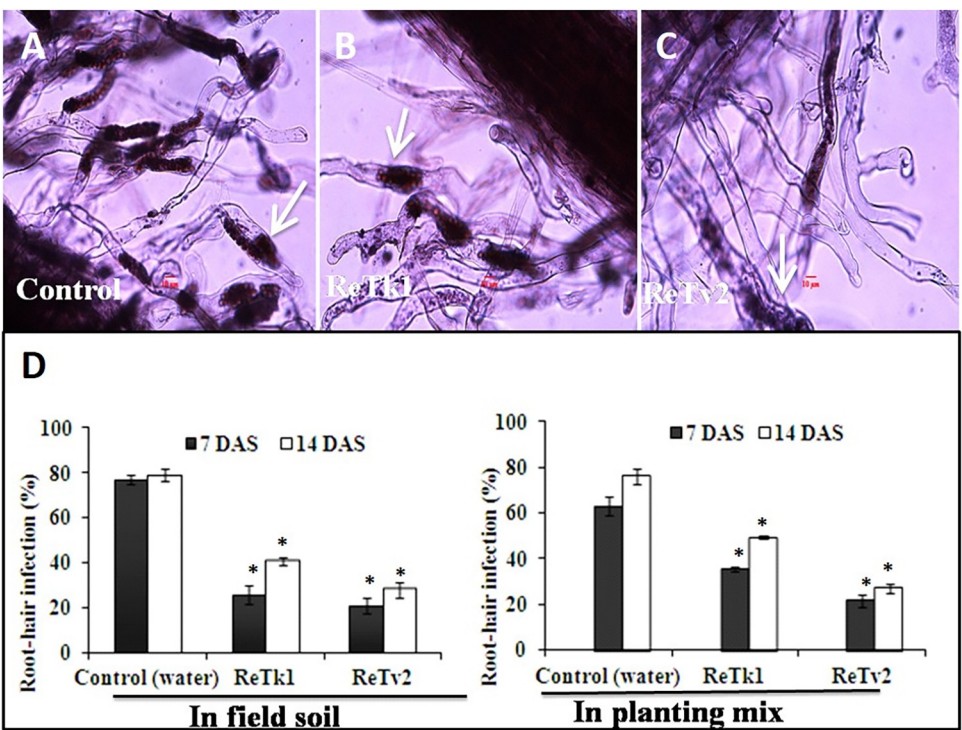

**Fig 3. Effect of *Trichoderma* spp. ReTk1 and ReTv2 on infection of rapeseed root hairs by *P. brassicae* at 7 and 14 days after seeding (DAS).** Roots of three treated plants were uprooted, washed and rinsed to remove most of the microbes, cut into 1 cm pieces, and fixed in 70% ethanol. Roots of control plants (pathogen-inoculated only) were processed similarly. Root pieces were stained overnight with aceto-carmine (1%) and examined using a light microscope for the presence of primary plasmodia or zoosporangia in root hairs (arrowhead). Root hairs of control plants showing damage due to infection by *P. brassicae* (A). Root haris of ReTk1 (B) and ReTv2-treated (C) plants showing less damage to the roots due to infection by *P. brassicae*. Bars indicate the average of six replications and the experiment repeated thrice (D). Capped lines represent standard error. The symbol "*" on top of the bar indicates a significant difference compared to the control at *P<0.05*.

to untreated controls (water-treated and pathogen-inoculated only) (Fig 3). Compared to controls, isolate ReTk1 and ReTv2 reduced root infection at 7 and 14 DAS, respectively, in field soil and planting mix. The clubroot incidence and severity in plants grown in both field soil and planting mix were also effectively reduced by treatments with the fungal isolates. In plants grown as untreated controls in field soil and planting mix, respectively, the clubroot root incidence was 85.00% and 89.10%; however, it was reduced to 57.10 and 47.70%, and 44.0 and 42.30% in plants grown in field soil and planting mix and treated with ReTk1 and ReTv2, respectively (Table 2). The clubroot root severity index in untreated controls grown in field soil and planting mix was 64.00 and 62.50%, respectively, but it was 29.30 and 28.00% and 23.00 and 21.00% in plants treated with ReTk1 and ReTv2, respectively (Table 3).

Rapeseed plants inoculated with *P. brassicae* and treated with ReTk1 and ReTv2 were also examined for plant growth and development. The shoots of untreated plants (only inoculated with the pathogen) were smaller than those treated with ReTk1 and ReTv2. The endophytes (ReTk1 and ReTv2) increased shoot length by 63 to 66% in field soil and 23 to 50% in planting mix compared to controls. Likewise, endophytes-treated plants had a 45–50% higher fresh shoot weight in planting mix and 46–60% higher fresh shoot weight in field soil than controls. ReTk1 and ReTv2 treatment of *P. brassicae*-inoculated rapeseed plants produced larger roots than untreated control roots (Fig 4). Compared to controls, ReTk1 and ReTv2 increased root length by 14 to 23% in field soil and by 33 to 71% in the planting mix. Following the treatment

**Table 2. Effect of *Trichoderma* spp. ReTk1 and ReTv2 on clubroot incidence and severity index of rapeseed in field soil and planting mix.**

| Treatment | Clubroot incidence | | Clubroot severity index | |
|---|---|---|---|---|
| | Field soil | Planting mix | Field soil | Planting mix |
| **Control (CK)** | 85.00±3.87a | 89.12±1.77a | 64.00±2.44a | 62.50±3.81a |
| **ReTk1** | 57.13±4.51b | 47.66±1.45b | 29.27±0.70b | 28.00±3.51b |
| **ReTv2** | 44.00±4.0c | 42.28±2.33b | 23.00±1.22c | 21.00±0.94b |
| **Significance level (*P*)** | *** | *** | *** | *** |

***P<0.001

Data are means ± standard error from a representative experiment. Each value represents the average of three replicates. Mean values with the same letters are not significantly different: Duncan's Multiple Range Test; *P<0.05*.

of *P. brassicae* inoculated plants with ReTk1 and ReTv2, more robust root systems with many lateral roots were produced, despite the fresh weight being less in some plants treated with ReTk1 and ReTv2. Fewer and smaller galls in the roots of the treated plants resulted in lower fresh root weight, whereas the untreated pathogen-inoculated rapeseed plants had more and larger galls with damaged roots (S2 Fig in S1 File). Consequently, in field soil and planting mix trials, *P. brassicae*-inoculated plants treated with ReTk1 and ReTv2 had a lower root index than the control (Fig 5). Gall development was negatively correlated with root index, with a lower root index indicating less gall development on shorter, weaker root systems following pathogen infection (S2 Fig in S1 File).

## Application timing of *Trichoderma* spp. and clubroot severity

An experiment was conducted to evaluate if using ReTk1 and ReTv2 during seeding, transplanting, or both seeding and transplanting were more effective in targeting *P. brassicae* zoospores. The endophytic fungal isolates ReTk1 and ReTv2 either applied at seeding, transplanting, or both significantly reduced the gall formation at 7 and 14 DAS compared to water-treated control (pathogen inoculated only) (Table 3). Equally, applying *Trichoderma* into field soil and planting mix effectively reduced the gall formation substantially. Similarly, endophytic fungal isolates ReTk1 and ReTv2 did not vary significantly in reducing gall

**Table 3. Effect of application timing of *Trichoderma* spp. ReTk1 and ReTv2 on clubroot severity index in field soil.**

| Inoculation/Treatment timing | | Mean clubroot severity index | | | |
|---|---|---|---|---|---|
| | | Field soil | | Planting mix | |
| At seeding | At transplanting | Transplanted at 7DAS | Transplanted at 14DAS | Transplanted at 7DAS | Transplanted at 14DAS |
| **Pb+Water** | **Water** | 61.05±1.45a* | 62.89.05±2.48a | 55.90±4.10a | 57.74±1.17a |
| **Pb+ReTk1** | **Water** | 21.87±3.12b | 17.03±2.97b | 18.54±0.90b | 19.68±2.18bc |
| **Pb+ReTv2** | **Water** | 14.75±1.42bc | 14.50±0.87b | 14.65±2.9b | 15.74±2.11bc |
| **Pb+ Water** | **Pb+ReTk1** | 18.03±2.41bc | 22.11±2.88b | 18.55±1.89b | 24.03±0.96b |
| **Pb+ Water** | **Pb+ReTv2** | 13.05±1.52c | 14.85±3.32b | 12.38±1.27b | 14.44±3.7bc |
| **Pb+ReTk1** | **Pb+ReTk1** | 15.41±2.21bc | 16.75±2.69b | 16.66±3.33b | 19.7±3.6bc |
| **Pb+ReTv2** | **Pb+ReTv2** | 10.83±2.5c | 13.45±1.92b | 10.20±0.20b | 11.32±0.21c |
| **Significance level (*P*)** | | *** | *** | *** | *** |

***P<0.001

*Data are means ± standard error from a representative experiment. Each value represents the average of three replicates. Mean values with the same letters are not significantly different: Duncan's Multiple Range Test; *P<0.05*.

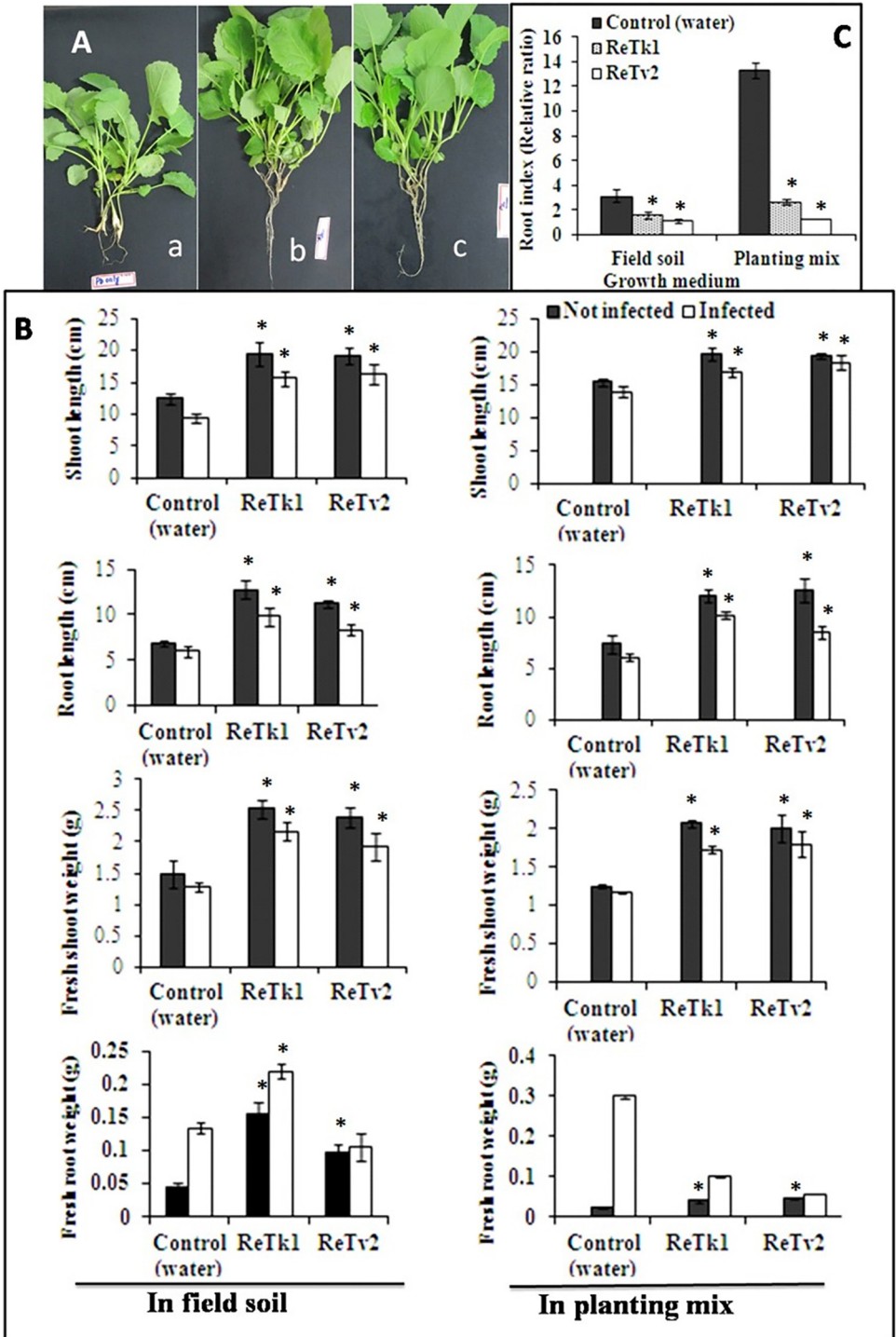

**Fig 4. Suppression of clubroot formation and promotion of plant growth of rapeseed.** The experiment was conducted with filed soil and planting mix under greenhouse conditions. The growth parameters and root index were quantified after 42 days of seeding. (A) Phenotypes of rapeseed plants without treatment (a) or treatment with ReTk1 (b) and ReTv2 (c). (B) Growth promotion of infected plants of rapeseed with or without *Trichoderma*-treatment and (C) root index. Root index means the fresh root weight of infected plants to the fresh root weight of uninfected plants ($R_i/R_{ni}$). The smaller root index means less gall development and the larger root index means more gall development and shorter root length during infection. Bars represent the average six replications and capped lines standard errors. The symbol "*" on top of the bar indicates a significant difference compared to the control at $P<0.05$.

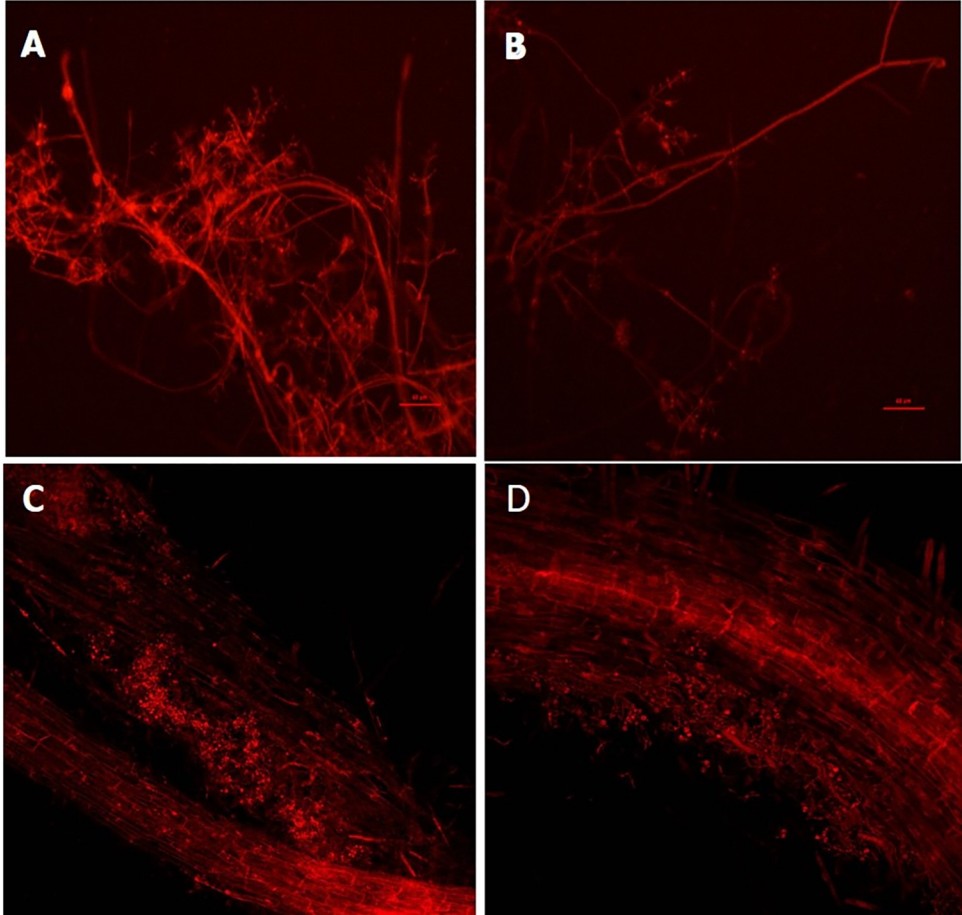

**Fig 5. Colonization of rapeseed roots by *Trichoderma* spp. ReTk1 and ReTv2.** *Trichoderma* transformant expressing *rfp*. (A & C) Root colonization by ReTktr-1(5). (B & D) Root colonization by ReTvtr-2(23).

formation when treated during seeding, transplanting, or both. However, the isolate ReTv2 was more effective in reducing clubroot severity than ReTk1 both in the field soil and soilless planting mix experiments (Table 3). The disease severity index was always greater than 61% in the untreated control (pathogen-inoculated only), whereas ReTk1 & ReTv2 treated plants in both experiments (field soil and planting mix experiments) showed disease severity indices below 30% (Table 3).

### *Trichoderma* colonization in the rapeseed roots

Rapeseed root inoculated with *Trichoderma* spp. ReTktr-1(5) and ReTvtr-2(23) showed profuse growth of the fungi. The fungi were observed in the root hairs and cortical tissues. Fluorescent microscopic observation also revealed the heavy presence of ReTktr-1(5) and ReTvtr-2(23) hyphae in the rapeseed roots. Both strains of *Trichoderma* were able to colonize the rapeseed rhizoplane and establish the hyphal growth inside the cortex of the root systems, indicating the signs of *Trichoderma* colonization of the rapeseed roots (Fig 5).

### Up-regulation of defence-related gene expression of rapeseed

Seven of the nine plant defense-related genes examined were up-regulated in ReTk1 and ReTv2-inoculated roots compared to control roots (Fig 6). The genes *BnOPCL* and *BnCCR*,

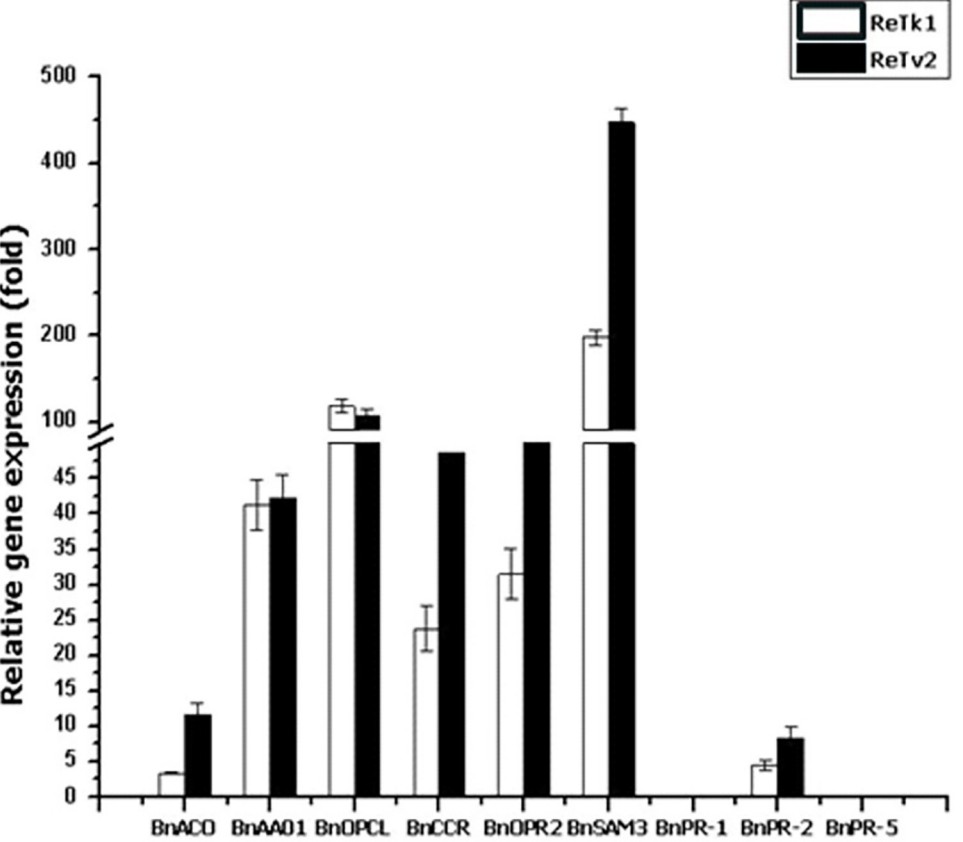

**Fig 6. Transcript level analysis of nine selected plant-defense-related genes in rapeseed seedlings treated with a soil drench of *Trichoderma* spp. or water (control) was performed using quantitative reverse-transcriptase polymerase chain reaction (qPCR).** The primers used in this experiment for quantifying *BnPR-1*, *BnPR-2* and *BnPR-5* were based on Potlakayala et al. [45] and the rest on Zhao et al. [46]. Gene expression was normalized using the reference *Actin* gene [46] in qPCR. The genes included in this experiment are three pathogenesis-related (*BnPR-1*, *BnPR-2* and *BnPR-5*), genes that control ethylene (*BnSAM3* and *BnACO*), auxin (*BnAA01*), jasmonic acid (*BnOPR2*), or phenylpropanoid (4-cournarate CoA ligase (*BnOPCL*) and c-innamoyl CoA reductase (*BnCCR*) pathways were assessed using root and first true leaf samples taken at 14 days after seeding (DAS). The relative levels of the transcript were calculated by the comparative Ct method. Bars represent the means and capped lines standard error (three replications). A gene was considered up-regulated when its expression level was substantially higher in treated plants relative to that of the control (least significant difference, $P < 0.05$).

encoding phenylpropanoid pathway enzymes, were up-regulated by 118.21-, 106.58- and 23.8-, 48.63-folds in the roots of ReTk1 and ReTv2-treated plants, respectively, relative to control. Ethylene signalling genes *BnACO* and *BnSAM3* increased by 3.41- and 11.75-folds and 197.5- and 447.12-folds in ReTk1 and ReTv2-treated roots, respectively. Likewise, the relative expression of *BnOPR2* involved in jasmonic acid pathways was enhanced by 31.5- and 54.84-folds in ReTk1 and ReTv2-inoculated roots, respectively, compared to the control. The auxin responsive, *BnAA01*, was up-regulated by 41.32-folds and 42.28-folds in the roots of ReTk1 and ReTv2-treated plants, respectively. The expression levels of the salicylic acid pathway gene *BnPR*-2 encoding pathogenesis-related protein were 4.5- and 8.34-fold greater in TeTk1- and ReTv2-treated roots, respectively, than in control roots. The relative expression level of these genes was consistently higher in ReTv2-treated roots than in ReTk1. In contrast, treatment with ReTk1 and ReTv2 did not affect the expression of two additional salicylic acid-sensitive PR-protein genes, *BnPR1* and *BnPR5* (Fig 6).

## Discussion

Most plants harbour a broad spectrum of endophytic fungi that act beneficially on plants by modulating host nutrition, growth, metabolites and stress responses [48–50]. The success of endophytic biocontrol agents largely depends on their ability to colonize the root or rhizospheric region, allowing them to thrive during the cropping period and work through their various biocontrol mechanisms against notorious plant pathogens. Our investigations demonstrate that the endophytic fungi *Trichoderma* spp. ReTk1 and ReTv2 isolated from healthy rapeseed roots (sampled from the clubroot-infested field) promote growth and reduce clubroot severity caused by an economically very important pathogen *P. brassicae*. The effects of *Trichoderma* on plant growth were noticeable in experiments conducted separately using field soil and soilless planting mix. The *Tichoderma*-treated plants produced larger shoots and roots compared to the untreated control. Likewise, leaf size and plant biomass production were significantly increased in the presence of the fungi, demonstrating the beneficial effects of these strains on plant performance. Previous work demonstrated that *Trichoderma* promotes growth responses in rapeseed [27, 51] and other plants [18, 31, 52, 53].

Although plant growth promotion by *Trichoderma* spp. has been studied extensively, little is known about their potential to inhibit clubroot fungus *P. brassicae*. In our experiment, rapeseed plants inoculated with *P. brassicae* and treated with the two endophytic isolates ReTk1 and ReTv2 also showed significantly enhanced growth compared to plants inoculated with *P. brassicae* only. The treatment of infected plants with ReTk1 and ReTv2 produced larger shoots and vigorous root systems with many lateral roots. Moreover, the percentage of root-hair infection (RHI) by *P. brassicae* was significantly reduced by ReTk1 and ReTv2 at 7 and 14 DAS, respectively, compared to pathogen inoculated only. In both field soil and planting mix experiments, the disease severity index was higher than 61% in the pathogen-inoculated control, while it was lower than 30% in the ReTk1 and ReTv2 treated plants. These *Trichoderma* strains were equally effective, whether applied at seeding, transplanting, or both. According to these findings, the endophytic strain *Trichoderma* spp. ReTk1 and ReTv2 have the potential to act as a biological control agent against rapeseed clubroot pathogen *P. brassicae*. The genus *Trichoderma* contains multiple species, many of which have been extensively researched for their potential use as biological control agents against a wide range of plant pathogens including *P. brassicae*. Experiments conducted in greenhouse pots revealed that *T. harzianum* strain T4 had a control efficiency of approximately 79% against *P. brassicae* in Chinese cabbage [28]. In another piece of research, the antifungal properties of the *T. harzianum* strain LTR-2 were demonstrated in Chinese cabbage grown outdoors. The disease incidence was reduced from 96.7% in the untreated control to 51.3% in the plants that had their seeds treated with spores of *T. harzianum* LTR-2 [29]. When tested against *P. brassicae* in Chinese cabbage, *T. harzianum* strain T4 showed up to 79% biocontrol efficacy in greenhouse pot trials [28]. Chinese cabbage disease incidence was reduced from 96.7% to 51.3% when *T. harzianum* strain LTR-2 was used as a biocontrol agent in the field [29]. Rapeseed seed germination and root growth were promoted by fermentation broth of *T. koningiopsis* Hz36, which also significantly reduced the viability of dormant *P. brassicae* spores [30]. The Hz36 strain was effective at controlling clubroot in rapeseed and *Arabidopsis thaliana* by 44.29% and 52.18%, respectively, in a biocontrol study. These findings suggest that clubroot pathogens can be controlled by using biological control strategies with *Trichoderma*.

The beneficial effects of these fungi are attributed to a number of direct and indirect mechanisms, such as mycoparasitism, antibiosis, the promotion of plant growth, and stress tolerance. *Trichoderma* treatment of *P. brassicae*-infected plants may result in the formation of new roots, allowing the continuous development of the plants and reducing the severity of clubroot [4]. *Trichoderma* might also produce a range of metabolites that could be detrimental to *P.*

*brassicae* [54]. In the present study, the two biocontrol strains, ReTk1 and ReTv2 exhibited good inhibitory effects on germination and viability of resting spores. These results suggest that ReTk1 and ReTv2 specifically target *P. brassicae* zoospores, the most vulnerable stage of the pathogen life cycle [55]. Primary and secondary zoospore production peaks between 7 and 14 days post-inoculation under optimal temperature conditions [43]. The application timing of the endophytic strains can be optimized at this stage to prevent root infection and colonization. Our investigation also suggests that root colonization by *Trichoderma* may play an integral role in protecting rapeseed plants against *P. brassicae* infection. The candidate competent *Trichoderma* strains are able to colonize plant roots without causing any damage to the plant tissues, but they do induce changes in the plant physiology and its defense system against the pathogen. Studies in the last decades have established that root colonization by *Trichoderma* isolates can induce resistance to various pathogens in different plants, both below and aboveground [56]. The fact that seven out of nine defense-related genes were expressed in *Trichoderma* colonized rapeseed roots in this study lends credence to the hypothesis that induced resistance may play a role in the *Trichoderma*-mediated suppression of clubroot disease on rapeseed. This systemic resistance is most likely the outcome of the regulation of the plant defense network, which may translate *Trichoderma*-induced early signaling events into a more efficient activation of defense responses. It is widely known that the phytohormones, jasmonic acid, salicylic acid, and ethylene function as key signals in the regulation of induced plant defense responses [14]. The present study shows the upregulation of jasmonic acid (*BnOPR2*), ethylene (*BnACO* and *BnSAM3*), phenylpropanoid (*BnOPCL* and *BnCCR*), auxin (*BnAAO1*) and salicylic acid (*BnPR2*) responsive defense genes in *Trichoderma* colonized roots. Classically, the SA-regulated signaling route is operated during pathogen-induced systemic acquired resistance (SAR) [57], whereas the jasmonic and ethylene pathways are typically necessary for the induction of systemic resistance by beneficial microbes [33, 58]. Therefore, the root endophytic *Trichoderma* spp. is capable of inducing resistance via SAR and ISR signaling pathways. Activation of multiple defense signaling pathways by plant growth-promoting fungi has been reported [14]. Moreover, the elevated expression of genes in phenylpropanoid pathways may lead to an enhanced amount of phenolic compounds [34], resulting in general resistance, including cell wall lignifications or specific resistance responses such as the production of phytoalexins [59].

## Conclusions

Overall, two new potential endophytic fungal agents, *Trichoderma* spp. ReTk1 and ReTv2, identified from the rapeseed rhizosphere, effectively promote growth and reduce the clubroot severity of rapeseed after being mixed into the soil. Both one and two treatment applications at seeding and transplanting 7 to 14 DAS were equally effective. The observed suppression of clubroot by *Trichoderma* spp. may involve ISR through the phenylpropanoid, JA/ET, and SA pathways. Therefore, *Trichoderma* isolates ReTk1 and ReTv2 have the potential to be utilized as biocontrol agents for managing clubroot and enhancing rapeseed growth.

## Supporting information

**S1 File.**
(DOCX)

## Acknowledgments

We would like to thank the authority of the University of Rajshahi and Huaxhong Agricultural University, PR China, for providing the necessary support to conduct this research.

## Author Contributions

**Conceptualization:** Mahmodol Hasan, Motaher Hossain, Daohong Jiang.

**Data curation:** Mahmodol Hasan.

**Formal analysis:** Mahmodol Hasan.

**Funding acquisition:** Motaher Hossain.

**Investigation:** Mahmodol Hasan, Motaher Hossain.

**Methodology:** Motaher Hossain, Daohong Jiang.

**Writing – original draft:** Mahmodol Hasan, Motaher Hossain.

**Writing – review & editing:** Mahmodol Hasan, Motaher Hossain, Daohong Jiang.

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
