## [Decision Letter · Decision Letter 0]

17 Apr 2023

PONE-D-23-08914New endophytic strains of Trichoderma promote growth and reduce clubroot severity of rapeseed (Brassica napus)PLOS ONE

Dear Dr. Hossain,

Thank you for submitting your manuscript to PLOS ONE. After careful consideration, we feel that it has merit but does not fully meet PLOS ONE’s publication criteria as it currently stands. Therefore, we invite you to submit a revised version of the manuscript that addresses the points raised during the review process.

We look forward to receiving your revised manuscript.

Kind regards,

Eugenio Llorens

Academic Editor

PLOS ONE

   "We would like to thank the authority of Huaxhong Agricultural University, PR China for providing necessary support to conduct this research. We are also greatfull to University of Rajshahi for partial financial support through allocation of special research grant (A-795/6/-109) (rsearch), 2022) to carry out this research."

Reviewers' comments:

Reviewer's Responses to Questions

**Comments to the Author**

1. Is the manuscript technically sound, and do the data support the conclusions?

Reviewer #1: Yes

Reviewer #2: Yes

2. Has the statistical analysis been performed appropriately and rigorously? 

Reviewer #1: Yes

Reviewer #2: Yes

3. Have the authors made all data underlying the findings in their manuscript fully available?

Reviewer #1: No

Reviewer #2: Yes

4. Is the manuscript presented in an intelligible fashion and written in standard English?

Reviewer #1: Yes

Reviewer #2: Yes

5. Review Comments to the Author

Reviewer #1: The manuscript entitled "New 1 endophytic strains of Trichoderma promote growth and reduce clubroot severity of rapeseed (Brassica napus)" describes the sampling of endophytic fungi from rapeseed and how two strains of Trichoderma are beneficial to the plant. The manuscript is well-written and uses an interesting approach to isolate the fungal strains directly from the plants. Here are some suggestions to improve the overall quality of the manuscript:

- Line 118: Fungal identification at the species level using only ITS sequences are not accurate, especially for Trichoderma strains, since several species share the same ITS sequence. I suggest using other loci, like rpb2 and tef1 (Cai & Druzhinina, 2021 - https://doi.org/10.1007/s13225-020-00464-4), or describing it in the paper as unconfirmed species.

- Send Table 1 to the supplemental material.

- Figure 1 shows very low support values on critical nodes. I recommend rerunning the tree using a better methodology like iqtree selecting the best model automatically.

- Table 2: Are these Genbank accession numbers from your sequences or the blast best hit on Genbank? If the latter is true, The authors must deposit the sequences in GenBank and add the accession numbers, and then the readers can replicate the same analysis. Also, the similarity search should be done on specialized databases, like UNITE (https://unite.ut.ee/)

- The numbers in the parenthesis on lines 345, 347, and 350 are confusing. Since the authors are already referring to Figure 2, remove them. Also, add the P-value in the figure for each comparison in ALL bar plots present in the manuscript. At least a "*" if < 0.05.

- Same on lines 387 and 389. It seems the same data shown in Tables 3 and 4 are also shown in Figures 4 and 5, is that correct? If so, remove the tables and include the statistical analysis in the bat plots.

- Increase the resolution of the plots. The quality is very low in the manuscript document.

- Both Trichoderma strains look good biocontrol candidates. In your opinion, how would they compare to commercial strains such as T. afroharzianum T22?

Reviewer #2: GENERAL ASPECTS:

- The document presents too many self-citations that are not scientifically justified. For example, the citation [18] is a book chapter and there are numerous papers on the subject that should be cited instead. For example: Poveda, J., Eugui, D., Abril-Urías, P., & Velasco, P. (2021). Endophytic fungi as direct plant growth promoters for sustainable agricultural production. Symbiosis, 85(1), 1-19. OR: J. Poveda, P. Baptista, S. Sacristán, P. Velasco

Beneficial effects of fungal endophytes in major agricultural crops. Front. Plant Sci., 13 (2022).

- Scientific names are named in full only the first time they appear in the text, then with the genus reduced. Example: B. napus.

- Review the function of the genes used in the gene expression study. The Abstract indicates that they are biosynthesis genes, but several genes that are response, not biosynthesis, have been used.

- Gene names are always in italics.

ABSTRACT:

- To point out the importance of rapeseed cultivation.

INTRODUCTION:

- To point out the importance of rapeseed cultivation.

- Develop a paragraph on the studies developed so far on the Trichoderma-B. napus interaction. There is a review published in 2022 that will be very useful for this: Poveda, J., Díaz-González, S., Díaz-Urbano, M., Velasco, P., & Sacristán, S. (2022). Fungal endophytes of Brassicaceae: Molecular interactions and crop benefits. Frontiers in Plant Science, 13, 932288. SEARCH THE TEXT FOR THE KEY WORDS: "rapeseed" and "napus".

MATERIALS AND METHODS:

- L118-137: the identification of Trichoderma based only on ITS does not allow to know the species, only the genus. For this, at least two other molecular markers are necessary. Correct this in all the work.

- L141: there is an error in "in the following section of this manuscript". The following section does not correspond to what is indicated.

- Data analysis: errors when speaking of P. Capitalize or lowercase, but always in italics.

RESULTS:

- L305-338: the fungal identification based only on ITS does not allow to know the species, only the genus. For this, at least two other molecular markers are necessary. Correct this in all the work.

DISCUSSION:

- Discuss further the reported in planta effects. There are many papers on Trichoderma-Rapeseed interaction and its effect on productivity, growth or ISR, which have not been discussed.

6. PLOS authors have the option to publish the peer review history of their article (what does this mean?). If published, this will include your full peer review and any attached files.

Reviewer #1: No

Reviewer #2: No

---

## [Author Response · Author response to Decision Letter 0]

31 May 2023

Responses to the Reviewer’s Comments

Reviewer #1: The manuscript entitled "New 1 endophytic strains of Trichoderma promote growth and reduce clubroot severity of rapeseed (Brassica napus)" describes the sampling of endophytic fungi from rapeseed and how two strains of Trichoderma are beneficial to the plant. The manuscript is well-written and uses an interesting approach to isolate the fungal strains directly from the plants. Here are some suggestions to improve the overall quality of the manuscript.

Response:

We thankfully acknowledge your overall positive comment on our manuscript. We are very glad that the Reviewer provided constructive comments and valuable suggestions that have helped us further improve the quality of our manuscript. Please find below our point-to-point responses to your kind comments and suggestions. We have used Word Track Changes to mark up the manuscripts with revisions. We hope that the Reviewer would be satisfied with our responses and endorse the revised manuscript for publication. 

Comment#1

Line 118: Fungal identification at the species level using only ITS sequences are not accurate, especially for Trichoderma strains, since several species share the same ITS sequence. I suggest using other loci, like rpb2 and tef1 (Cai & Druzhinina, 2021 - https://doi.org/10.1007/s13225-020-00464-4), or describing it in the paper as unconfirmed species. 

Response:

Thank you for your suggestion. The species-level identification has been omitted and only Trichoderma spp. (ReTk1 and ReTv2) have been used throughout the manuscript. Moreover, we have added sentences describing the need to use additional loci for species-level identification of the two strains citing the aboved reference (L454-459) . 

Comment#2

Send Table 1 to the supplemental material.

Response:

Table 1 has been transferred to supplemental material as Table S1.

Comment#3

Figure 1 shows very low support values on critical nodes. I recommend rerunning the tree using a better methodology like iq-tree, selecting the best model automatically.

Response:

Thank you for the valuable comments. We agree with you that IQ-TREE is a better methodology. However, we are not experts in using IQ-TREE. Therefore, we reanalyzed the tree using the latest version (v.11) of MEGA and cited a reference supporting this analysis in the Materials and Methods (L227). The new Figure 1 has been added to the manuscript.

Comment#4

Table 2: Are these Genbank accession numbers from your sequences or the blast best hit on Genbank? If the latter is true, The authors must deposit the sequences in GenBank and add the accession numbers, and then the readers can replicate the same analysis. Also, the similarity search should be done on specialized databases, like UNITE (https://unite.ut.ee/)

Response:

Thank you very much for this suggestion. The GenBank accession numbers of our sequences have been added to Table 1. The similarity search has been done using BLASTn databases.

Comment#5

The numbers in the parenthesis on lines 345, 347, and 350 are confusing. Since the authors are already referring to Figure 2, remove them. Also, add the P-value in the figure for each comparison in ALL bar plots present in the manuscript. At least a "*" if < 0.05.

Response:

Thank you for your suggestion. The values in the parenthesis have been removed. The P-value has been added to the bar plots in all figures.

Comment# 6

Same on lines 387 and 389. It seems the same data shown in Tables 3 and 4 are also shown in Figures 4 and 5; is that correct? If so, remove the tables and include the statistical analysis in the bat plots.

Response: 

Thank you for your suggestion. The values indicating the percent increase have been removed. However, Table 3 and Figure 4 and Table 4 and Figure 5 are not showing the same data. Table 3 represents the treatment effects on clubroot incidence and severity index of rapeseed in field soil and planting mix. In contrast, Table 4 denotes the application timing of Trichoderma spp. on clubroot severity index in field soil and planting mix. Figure 4 represents the root hair infection and Figure 5 represents the treatment effect on suppression of club root formation. 

Comment# 7

Increase the resolution of the plots. The quality is very low in the manuscript document.

Response:

The resolution of the plots has been enhanced. So, the quality is now improved. 

Comment#8

Both Trichoderma strains look good biocontrol candidates. In your opinion, how would they compare to commercial strains such as T. afroharzianum T22?

Response: 

Thank you for your valuable statement. Trichoderma spp. are well-known biocontrol agents and widely used worldwide, although their effectiveness varies from strain to strain. In our study, we isolated two strains that show good biocontrol and growth promotion activity. As T. afroharzianum T22 is a commercial strain, it is obviously a promising biocontrol agent. As per our opinion, the isolated two strains and T22 could be good candidate biocontrol agents for reducing clubroot severity and enhancing the growth of rapeseed. 

Reviewer #2: 

Thank you very much for your positive evaluation of our manuscript. We also very much appreciate your overall comments and suggestions that have helped us improve the quality of our manuscript. Please see below our point-to-point responses to your kind comments and suggestions. Word Track Changes have been used to note up revisions in the manuscripts. We hope that the Reviewer would be satisfied with our responses and endorse the revised manuscript for publication. 

Comment#1

GENERALASPECTS:

The document presents too many self-citations that are not scientifically justified. For example, the citation [18] is a book chapter and there are numerous papers on the subject that should be cited instead. For example: Poveda, J., Eugui, D., Abril-Urías, P., & Velasco, P. (2021). Endophytic fungi as direct plant growth promoters for sustainable agricultural production. Symbiosis, 85(1), 1-19. OR: J. Poveda, P. Baptista, S. Sacristán, P. Velasco. Beneficial effects of fungal endophytes in major agricultural crops. Front. Plant Sci., 13 (2022). 

Response: 

We thank the Reviewer for making this suggestion. Following your advice, we have replaced and added new references to the manuscript (L104).

Comment#2

Scientific names are named in full only the first time they appear in the text, then with the genus reduced. Example: B. napus.

Response:

Thank you for critical reviewing. The scientific name has been corrected in the manuscript according to your suggestion. 

Comment#3

Review the function of the genes used in the gene expression study. The Abstract indicates that they are biosynthesis genes, but several genes that are response, not biosynthesis, have been used.

Response:

Thank you for your suggestion. We have corrected the sentence in the Abstract (L42) following your suggestion.

Comment 4

Gene names are always in italics.

Response:

All the gene names have been written in italics.

Comment#5

ABSTRACT:

To point out the importance of rapeseed cultivation.

Response:

A sentence describing the importance of rapeseed cultivation has been added in the Abstract (L24-25).

Comment#5

INTRODUCTION:

To point out the importance of rapeseed cultivation.

Response:

The importance of rapeseed cultivation has been described in Introduction (L62-65).

Comment#6

Develop a paragraph on the studies developed so far on the Trichoderma-B. napus interaction. There is a review published in 2022 that will be very useful for this: Poveda, J., Díaz-González, S., Díaz-Urbano, M., Velasco, P., & Sacristán, S. (2022). Fungal endophytes of Brassicaceae: Molecular interactions and crop benefits. Frontiers in Plant Science, 13, 932288. SEARCH THE TEXT FOR THE KEYWORDS: "rapeseed" and "napus". 

Response:

We appreciate your suggestion. A paragraph on the discussion of Trichoderma-B. napus interaction has been added in the Introduction (L104-164). 

Comment 7

MATERIALS AND METHODS: 

L118-137: Identifying Trichoderma based only on ITS does not allow us to know the species, only the genus. For this, at least two other molecular markers are necessary. Correct this in all the work.

Response:

We have accepted your suggestion and identified the fungal strains only at the genus level, such as Trichoderma spp. ReTk1 and ReTV2 (L455-460).

Comment 8

 L141: there is an error in "in the following section of this manuscript". The following section does not correspond to what is indicated.

Response:

We have corrected the statement as “described in the later section of this manuscript” (From L232-233). The method has been described

Comment 8

Data analysis: errors when speaking of P. Capitalize or lowercase, but always in italics.

Response: 

Now this has been italicized (p). 

Comment 9

RESULTS:

L305-338: the fungal identification based only on ITS does not allow to know the species, only the genus. For this, at least two other molecular markers are necessary. Correct this in all the work.

Response: 

We have identified the fungi at the species such as only Trichoderma spp. ReTk1 and ReTV2 and corrected it thoughout the manuscript.

Comment#10

DISCUSSION:

Discuss further the reported in planta effects. There are many papers on Trichoderma-Rapeseed interaction and its effect on productivity, growth or ISR, which have not been discussed.

Response:

A paragraph on Trichoderma-B. napus interaction has already been added in the “Introduction” section following your suggestion (L104-164).

---

## [Decision Letter · Decision Letter 1]

6 Jun 2023

PONE-D-23-08914R1New endophytic strains of Trichoderma promote growth and reduce clubroot severity of rapeseed (Brassica napus)PLOS ONE

Dear Dr. Hossain,

Thank you for submitting your manuscript to PLOS ONE. After careful consideration, we feel that the manuscript could be accepted for publication after the small corrections suggested by one of the reviewers. Therefore, we invite you to submit a revised version of the manuscript that addresses the points raised during the review process.

We look forward to receiving your revised manuscript.

Kind regards,

Eugenio Llorens

Academic Editor

PLOS ONE

Journal Requirements:

Reviewers' comments:

Reviewer's Responses to Questions

**Comments to the Author**

1. If the authors have adequately addressed your comments raised in a previous round of review and you feel that this manuscript is now acceptable for publication, you may indicate that here to bypass the “Comments to the Author” section, enter your conflict of interest statement in the “Confidential to Editor” section, and submit your "Accept" recommendation.

Reviewer #1: All comments have been addressed

Reviewer #2: All comments have been addressed

2. Is the manuscript technically sound, and do the data support the conclusions?

Reviewer #1: Yes

Reviewer #2: Yes

3. Has the statistical analysis been performed appropriately and rigorously? 

Reviewer #1: Yes

Reviewer #2: Yes

4. Have the authors made all data underlying the findings in their manuscript fully available?

Reviewer #1: Yes

Reviewer #2: Yes

5. Is the manuscript presented in an intelligible fashion and written in standard English?

Reviewer #1: Yes

Reviewer #2: Yes

6. Review Comments to the Author

Reviewer #1: The manuscript was clearly improved, but I still have two recommendations:

- Drop the Figure 1 with the phylogenetic tree, the support values are too low and the grouping seems random and uninformative. Therefore, it's not adding relevant information to the manuscript.

- Regarding the "*" on figures 2B,3C,4 and 5B: is not clear what it means. Just add a "*" on top of the bar where the comparison with the control is significant. The way it's shown, there's no way to know which comparison is significant.

Reviewer #2: Thank you for making all the requested changes. I believe that the document has been substantially improved and is suitable for publication. Congratulations for the work done and much success in your future work.

7. PLOS authors have the option to publish the peer review history of their article (what does this mean?). If published, this will include your full peer review and any attached files.

Reviewer #1: No

Reviewer #2: No

---

## [Author Response · Author response to Decision Letter 1]

11 Jun 2023

Responses to the Reviewer’s Comments

Reviewer #1: 

We thankfully acknowledge your overall positive comment on our manuscript. We are very glad that the Reviewer provided constructive comments and valuable suggestions that have helped us further improve the quality of our manuscript. Please find below our point-to-point responses to your kind comments and suggestions. We have used Word Track Changes to mark up the manuscripts with revisions. We hope that the Reviewer would be satisfied with our responses and endorse the revised manuscript for publication. 

Comment 1

Drop the Figure 1 with the phylogenetic tree, the support values are too low and the grouping seems random and uninformative. Therefore, it's not adding relevant information to the manuscript.. 

Response

Thank you for your suggestion. We have droped the Figure 1. 

Comment 2

 Regarding the "*" on figures 2B,3C,4 and 5B: is not clear what it means. Just add a "*" on top of the bar where the comparison with the control is significant. The way it's shown, there's no way to know which comparison is significant.

Response

We have amended Figures 2B, 3C, 4 and 5B following your suggestion.

---

## [Decision Letter · Decision Letter 2]

15 Jun 2023

New endophytic strains of Trichoderma promote growth and reduce clubroot severity of rapeseed (Brassica napus)

PONE-D-23-08914R2

Dear Dr. Hossain,

We’re pleased to inform you that your manuscript has been judged scientifically suitable for publication and will be formally accepted for publication once it meets all outstanding technical requirements.

Kind regards,

Eugenio Llorens

Academic Editor

PLOS ONE

Additional Editor Comments (optional):

Reviewers' comments:

Reviewer's Responses to Questions

**Comments to the Author**

1. If the authors have adequately addressed your comments raised in a previous round of review and you feel that this manuscript is now acceptable for publication, you may indicate that here to bypass the “Comments to the Author” section, enter your conflict of interest statement in the “Confidential to Editor” section, and submit your "Accept" recommendation.

Reviewer #1: All comments have been addressed

2. Is the manuscript technically sound, and do the data support the conclusions?

Reviewer #1: Yes

3. Has the statistical analysis been performed appropriately and rigorously? 

Reviewer #1: Yes

4. Have the authors made all data underlying the findings in their manuscript fully available?

Reviewer #1: Yes

5. Is the manuscript presented in an intelligible fashion and written in standard English?

Reviewer #1: Yes

6. Review Comments to the Author

Reviewer #1: The authors addressed all comments and removed the problematic figure. Now it's ready for publication.

7. PLOS authors have the option to publish the peer review history of their article (what does this mean?). If published, this will include your full peer review and any attached files.

Reviewer #1: No

---

## [Editor Report · Acceptance letter]

20 Jun 2023

PONE-D-23-08914R2 

New endophytic strains of Trichoderma promote growth and reduce clubroot severity of rapeseed (*Brassica napus*) 

Dear Dr. Hossain:

I'm pleased to inform you that your manuscript has been deemed suitable for publication in PLOS ONE. Congratulations! Your manuscript is now with our production department. 

Kind regards, 

on behalf of

Dr. Eugenio Llorens 

Academic Editor

PLOS ONE